# Twister ribozymes as highly versatile expression platforms for artificial riboswitches

Michele Felletti[1,2], Julia Stifel[1,2], Lena A. Wurmthaler[1,2], Sophie Geiger[1] & Jörg S. Hartig[1,2]

The utilization of ribozyme-based synthetic switches in biotechnology has many advantages such as an increased robustness due to *in cis* regulation, small coding space and a high degree of modularity. The report of small endonucleolytic twister ribozymes provides new opportunities for the development of advanced tools for engineering synthetic genetic switches. Here we show that the twister ribozyme is distinguished as an outstandingly flexible expression platform, which in conjugation with three different aptamer domains, enables the construction of many different one- and two-input regulators of gene expression in both bacteria and yeast. Besides important implications in biotechnology and synthetic biology, the observed versatility in artificial genetic control set-ups hints at possible natural roles of this widespread ribozyme class.

[1] Department of Chemistry, University of Konstanz, Universitätsstraße 10, 78457 Konstanz, Germany. [2] Konstanz Research School Chemical Biology (Kors-CB), University of Konstanz, Universitätsstraße 10, 78457 Konstanz, Germany. Correspondence and requests for materials should be addressed to J.S.H. (email: joerg.hartig@uni-konstanz.de).

The family of small endonucleolytic ribozymes is composed of RNA motifs of 50–150 nucleotides (nt) length with intrinsic RNA cleavage activity[1]. Until recently, only five classes of small trans-esterification ribozymes have been described: the hammerhead, the hairpin, the hepatitis delta virus, the Varkud satellite and the *glmS* ribozymes[1]. Recently, new widespread classes of self-cleaving ribozymes called twister, twister sister, pistol and hatchet were discovered using a bioinformatics pipeline[2,3]. Meanwhile, crystal structures of twister ribozymes have revealed a highly compact fold comprising two pseudoknots and have provided new insights into the catalytic mechanism[4–7]. Although physiological functions remain unknown, genetic contexts similar to hammerhead ribozymes (HHRs) were observed[2,3]. Recently it was shown that the twister ribozyme is able to activate the human RNA-activated protein kinase that is an innate immune signalling protein[8]. The discovery of these new motifs not only contributes to the extent of catalytic RNA biology, but also offers new tools to synthetic biologists for constructing artificial genetic switches. The utilization of ribozyme-based synthetic switches in biotechnology and future therapeutic applications has many advantages such as an increased robustness due to *in cis* regulation, small coding space and a high degree of modularity[9]. Ligand-dependent ribozymes can be engineered by attaching small molecule-sensing aptamer domains to the catalytically active scaffold[10]. In this regard, the naturally occurring *glmS* riboswitch and the bacterial cyclic-di-GMP-dependent group I intron represent pivotal examples as they exert their catalytic activities (self-cleavage and self-splicing, respectively) in a ligand-dependent way[11,12]. An artificial riboswitch based on the HHR was first introduced using a novel design in which the ribozyme motif sequesters the Shine–Dalgarno (SD) sequence enabling control of translation initiation in bacteria[13]. This system mimics the mode of action of some naturally occurring riboswitches[13]. The use of ribozymes in artificial RNA switches turned out to be very modular with respect to ligand specificity, type of regulated RNA, as well as the host organism[14–19]. An interesting application is the construction of RNA switches that sense two small molecular inputs, enabling the computing of Boolean logics. Using a rational design strategy, a NOR logic gate based on a hepatitis delta virus ribozyme in mammalian cells was generated[16]. We developed a genetic circuit that performs Boolean logics computation in *Escherichia coli* by combining orthogonal HHR switches for both transfer RNA (tRNA) and messenger RNA (mRNA) regulation[20]. In addition, HHR-based two-input Boolean operators have been reported in yeast[21]. However, the use of HHRs as a platform for two-input riboswitches requires stem I/II interactions[22].

Here we present for the first time a novel strategy that employs the twister ribozyme as an expression platform in artificial riboswitches. In particular, we show that the twister ribozyme is an extraordinarily versatile expression platform that allows the connection of three different aptamer domains at two independent positions (P1 and P5), generating a variety of one-input RNA switches that can act as efficient synthetic genetic regulators in both *E. coli* and yeast. Using a design that is mimicking the situation observed in some naturally occurring twister ribozymes where multiple optional domains are directly connected to the catalytic core in position P1 and P5 (ref. 2), we developed a series of compact two-input riboswitches that sense and respond to two small molecular signals at once (theophylline and thiamin pyrophosphate (TPP)) and that behave as Boolean logic operators in *E. coli*.

Thanks to the diversity of the obtained responses, our novel twister-based riboregulators have the potential of being used in a variety of biotechnological applications. Moreover their structural organization suggests regulatory functions for the variable domains observed in positions P1 and P5 in several naturally occurring twister ribozymes.

## Results

**The twister expression platform**. We based our synthetic riboswitch design on a naturally occurring type P3 twister motif which was identified in an environmental sequence (env-9) whose crystal structure was determined (Fig. 1a)[2,5]. The P3 class is the least represented type of twister ribozyme formats, with only nine motifs found in environmental sequences[2]. To develop switches in *E. coli* we used an *in vivo* screening strategy based on the masking of the SD sequence. This approach was demonstrated to be extremely effective for the generation of artificial riboswitches based on the HHR in *E. coli*[23]. We first tested the potential of the ribozyme as gene switches by inserting the motifs into the 5′-UTR of an enhanced green fluorescent protein (eGFP) reporter mRNA in two configurations (Fig. 1b–f and Supplementary Fig. 1). The SD sequence was placed and sequestered either in stem P1 or P3. Cleavage-inactive controls were generated by inverting two conserved nucleotides involved in the formation of the conserved stem P4 (Fig. 1d,e). Active twister ribozyme resulted in enhanced reporter gene activity in comparison to inactive ribozymes (5-fold in case of P1 and ∼100-fold for P3-format twister constructs, see Fig. 1g). The absolute eGFP expression of the active P3 construct is comparable to the positive control, which lacks any sequestration of the ribosome binding site and presents an unstructured 5′-UTR. On the contrary the active P1 construct shows much lower levels of eGFP expression when compared with the positive control. In the P1 construct a consistent secondary structure is still present in the 5′-UTR immediately upstream the SD sequence upon the cleavage. The presence of stem-loop structures in the so-called standby site (that is, the initial landing pad on the mRNA that is recognized by the 30S ribosomal subunit)[24] was shown to strongly inhibit translation especially if no single-stranded RNA element is present in the proximal region upstream the SD sequence[25]. For these reasons the naturally occurring P3 configuration was considered to be more suited for artificial riboswitch construction.

To prove that the differential gene expression in the active and inactive P3 construct is due to the sequestration of the SD sequence and not to the simple cleavage of the 5′-UTR of the eGFP gene, we compared the P3 SD format with two constructs where the active and the inactive catalytic env-9 motif is present in the 5′-UTR but in contrast the SD sequence is always accessible (Fig. 1f and Supplementary Fig. 1). Both constructs show eGFP expression comparable to the positive control (Fig. 1g), indicating that the cleavage activity in the 5′-UTR itself does not have an effect on the levels of eGFP expression. In addition, similar levels of eGFP mRNA were measured by reverse transcription semiquantitative polymerase chain reaction (RT-qPCR; Supplementary Fig. 2 and Supplementary Note 1). These data demonstrate that the ribozyme exerts its effect via controlling the accessibility of the Shine–Dalgarno region. Moreover, it shows for the first time that twister ribozyme-mediated cleavage activity is able to strongly affect protein expression.

**Twister-based artificial riboswitches in *Escherichia coli*.** We next generated ligand-dependent twister switches by attaching aptamer domains to additional sites of the catalytic domain of the P3-type twister construct (Supplementary Fig. 3). Although the core motif possesses a highly compact RNA fold, it contains several sites potentially suited for attaching ligand-sensing aptamer domains[2]. For considerations regarding the position of branch points, occurrence in natural motifs and their suitability

for designing ligand-dependent twister ribozymes see Supplementary Note 2. We chose positions P1 and P5 for attaching an artificial theophylline aptamer and a naturally occurring TPP binding motif as sensor domains[26,27]. In order to identify efficient RNA switches, clonal libraries composed of randomized nucleotides in the connection region between catalytic and sensor domains (so-called communication modules[28]) were screened for ligand-induced changes of gene expression (Fig. 2a and Supplementary Figs 4 and 5). Clones showing pronounced switching activity were further analysed. We were able to isolate many different artificial riboswitches that displayed changes in gene expression upon addition of the respective ligand to the growth medium. In Fig. 2b, twister-based switches carrying the theophylline aptamer in positions P1 and P5 are shown. Both increased (on-switches) as well as decreased (off-switches) expression levels upon theophylline addition were obtained, with the best switching performances (defined as ratio of fluorescence of active divided by inactive expression state) ~6-fold in case of the on-switch and ~10-fold for the off-switch (9-fold if calculated without background subtraction—Supplementary Fig. 4d).

TPP-dependent switches are shown in Fig. 2c, with the best off-switches observed with the TPP aptamer connected to position P1 (~35-fold inactivation with background subtraction and 16-fold without background subtraction–Supplementary Fig. 5d) and only moderate performance of on-switches (up to 5-fold). With regard to the connection site of the aptamer, in position P5 only off-switches were obtained and in general the switching efficiency appears to be lower compared with the twister motifs modified in position P1. Interestingly most of the identified communication sequences of the off-switches are composed of canonical base pairs (Supplementary Figs 4b and

5b). Cleavage-inactive control sequences generated for each described artificial riboswitch shut off gene expression completely (Supplementary Figs 4c and 5c), demonstrating that the catalytic activity of the twister motif is a prerequisite for acting as artificial riboswitches. To investigate whether the artificial riboswitches work at the single-cell level, we further characterized all riboswitches presented in Fig. 2b,c and the relative catalytically inactive mutants by flow cytometry (FC; Fig. 2d,e, Supplementary Figs 6–8 and Supplementary Table 1).

In addition, we determined kinetics of isolated ligand-dependent twister ribozymes. *In vitro* transcribed theophylline- and TTP-dependent off- and on-switches were incubated in presence and in absence of the ligand. Respective activation or inhibition of ribozyme cleavage activity was observed for all investigated aptazymes following ligand addition (Supplementary Fig. 9, Supplementary Tables 2 and 3 and Supplementary Note 3).

**Twister-based artificial riboswitches in yeast.** To demonstrate the versatility of the twister ribozyme as modular expression platform, we next aimed at the development of synthetic riboswitches in eukaryotic cells. Artificial aptazyme-based riboswitches located in the 3′-UTR of the gene of interest were already developed in yeast by different groups[29–31]. In eukaryotes the removal of the poly(A) tail upon cleavage reduces the stability of the mRNA (Supplementary Fig. 10). An engineered form of the env-9 motif was inserted into the 3′-UTR of the gene encoding for the GAL4 transcription factor in yeast. GAL4 activates the transcription of a genomically encoded *lacZ* gene that is used during the screening as a reporter (see Methods section for details about the expression system). A neomycin aptamer described earlier[32] was attached to the stem P1 using a communication module library of four randomized nucleotides (Fig. 2f). The

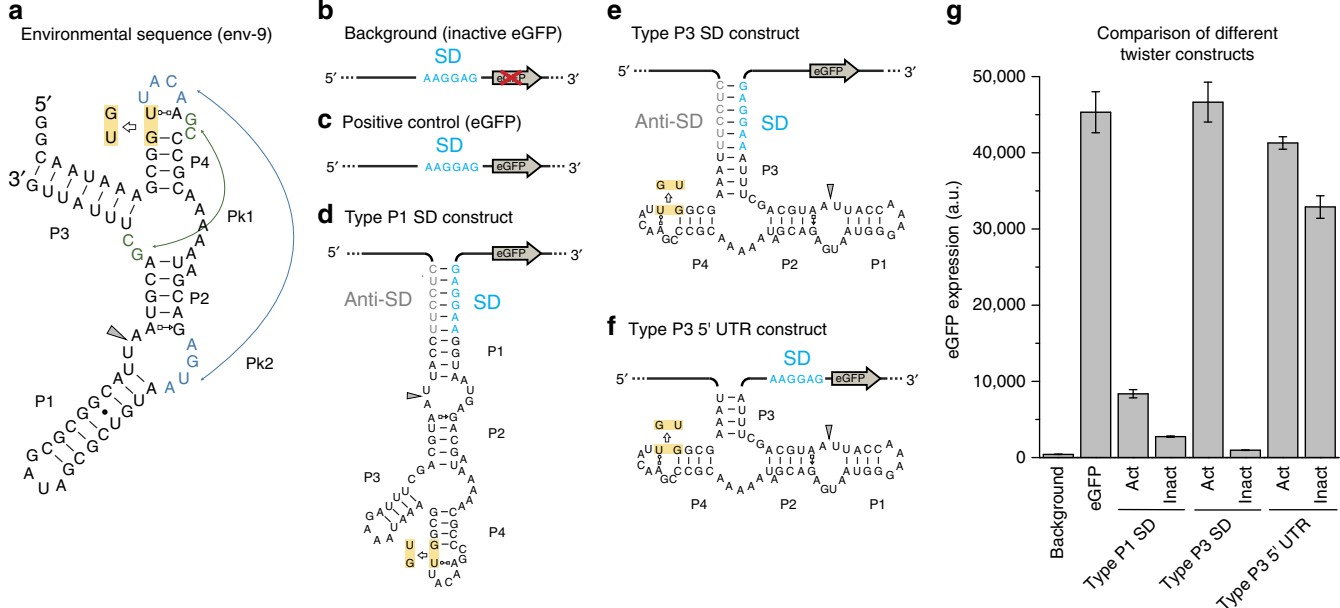

**Figure 1 | The env-9 twister ribozyme motif and its engineered 5′-UTR constructs in *Escherichia coli*.** The env-9 motif can be used to control gene expression at the translational level in E. coli. (**a**) Sequence and secondary structure of the natural occurring env-9 motif. The pseudoknots Pk1 and Pk2 are highlighted in green and blue, respectively. The inactivating mutation is highlighted in yellow. The cleavage site is indicated with a grey arrow. A negative control carrying a truncated form of eGFP (**b**) was used to measure the background fluorescence of the culture. A plasmid carrying the active form of eGFP (**c**) was used as a positive control. The Shine–Dalgarno sequence (SD) is shown in light blue. (**d**–**f**) The three engineered constructs inserted in the 5′-UTR of the eGFP gene. In the type P1 SD construct (**d**) the SD (in blue) and the anti-SD (in grey) sequences were included into stem P1 of the twister. In the type P3 SD construct (**e**) the SD and the anti-SD sequence were placed into stem P3. In the type P3 5′-UTR construct (**f**) the twister motif is inserted in the 5′-UTR before the SD of the eGFP gene (see Supplementary Fig. 1 for details). In the constructs shown in **e**,**f** the stem P1 was shortened and closed with a 5′-GAAA-3′ tetraloop. (**g**) The levels of eGFP expression (bulk fluorescence divided by the relative $OD_{600}$) of the constructs displayed in **b**–**f** are shown. The error bars represent s.d. calculated on independent biological triplicates.

library was screened directly in yeast for differential expression of the reporter gene in the presence and in the absence of neomycin in yeast culture medium (for details see Supplementary Note 4). The screening resulted in the isolation of two off-switches (Fig. 2g) with switching performances of ∼10-fold (Supplementary Fig. 11). Interestingly the two isolated communication modules contain only purine nucleotides (Supplementary Fig. 11b). The use of neomycin as an inducer in a eukaryotic model organism is relevant because it is an example of a compound already approved for therapeutic applications. The full sequences of all the one-input twister-based riboswitches in *E. coli* and yeast are shown in the Supplementary Table 4.

**Two-input twister-based artificial riboswitches in *Escherichia coli*.** The results described so far demonstrate that the twister ribozyme scaffold offers at least two independent attachment sites for ligand-sensing aptamer domains. This opens up the possibility of constructing compact two-input genetic switches that sense and

respond to two small molecular signals at once. We started from the P1- or P5-based theophylline and TPP riboswitches and sampled libraries of randomized communication modules in the second position with the second aptamer domain attached (Fig. 3, Supplementary Fig. 12 and Supplementary Note 5). We identified a broad range of switching behaviours which can be represented using binary Boolean logic gates such as AND, NAND, OR, NOR and ANDNOT (Fig. 3b,c). Depending on where the threshold is set, some switches can be assigned to one or to another category. Some switches show dependency only on one of the two ligands and can be described as buffer or inverter gates (Supplementary Figs 13–19). The flow cytometry characterization of some of the two-input riboswitches demonstrated that they work at the single-cell level (Fig. 3, Supplementary Figs 20–21 and Supplementary Table 5).

In addition, we validated the observed ligand-induced gene expression changes of some of the one- and two-input riboswitches utilizing a bioluminescence reporter assay by

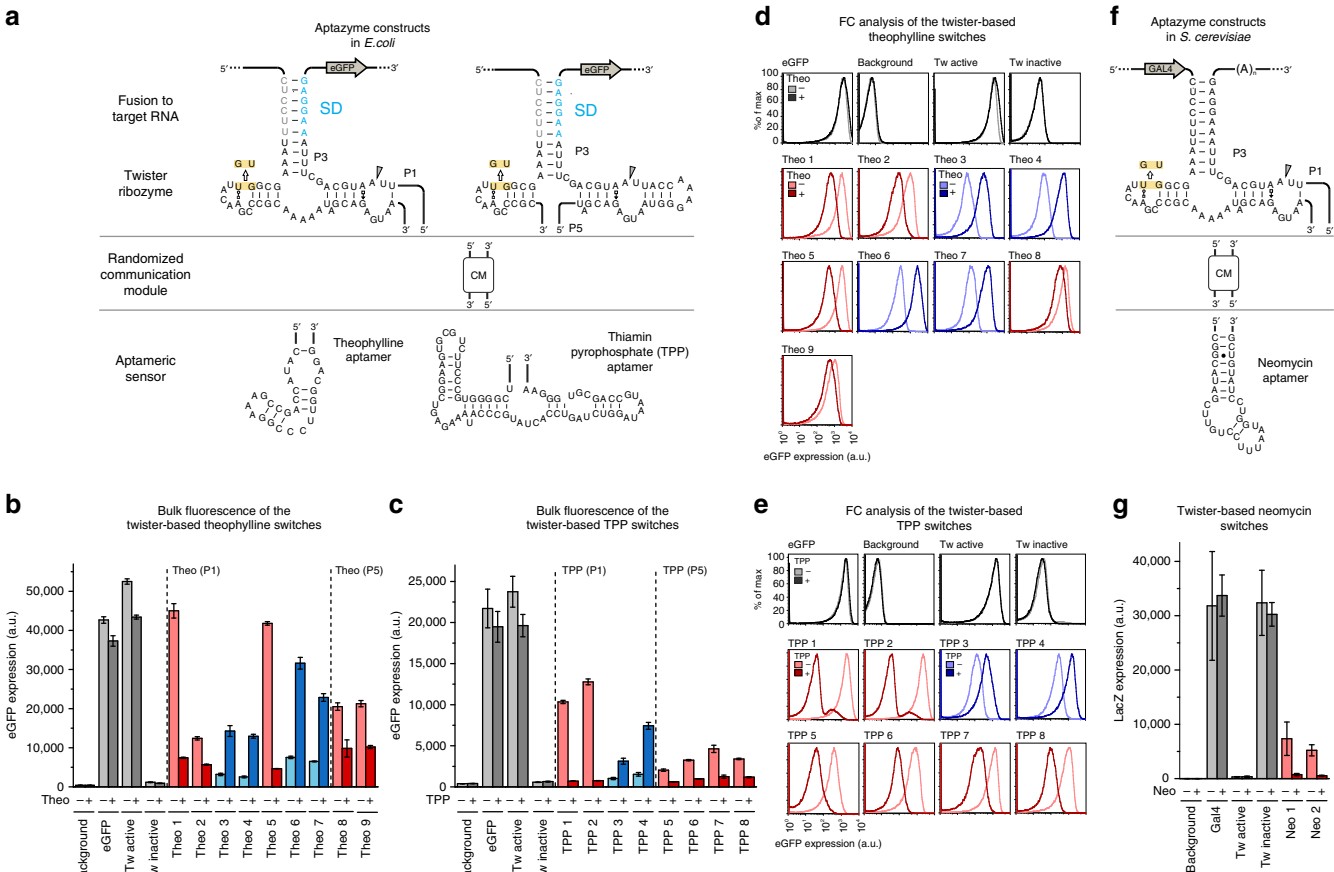

**Figure 2 | Twister-based one-input riboswitches in *Escherichia coli* and *Saccharomyces cerevisiae*.** (**a**) Theophylline and TPP aptazyme design in *E. coli*. Different types of communication modules (CM) were employed (see Supplementary Figs 4 and 5). The Shine-Dalgarno (SD) and the anti-SD are shown in light blue and grey, respectively. The inactivating mutation is highlighted in yellow. The cleavage site is indicated with a grey arrow. (**b,c**) Level of eGFP expression (bulk fluorescence divided by the relative $OD_{600}$) of the selected clones in the absence ( − ) and in the presence ( + ) of (**b**) 2.5 mM theophylline and (**c**) 1 mM thiamin in the culture medium. The levels of reporter gene expression in the different conditions are represented by grey tone bars for the controls, red tone bars for the off-switches and blue tone bars for the on-switches. The error bars represent s.d. calculated on independent biological triplicates. (**d,e**) flow cytometry (FC) histograms of (**d**) the theophylline switches and (**e**) the TPP switches recorded in the presence ( + ) and in the absence ( − ) of the respective ligands (2.5 mM theophylline and 1 mM thiamin in the culture medium respectively). The FC diagrams of the controls are represented by grey tone traces, the diagrams of the off-switches by red tone traces and the ones of the on-switches by blue tone traces. (**f**) Neomycin aptazyme design in *S. cerevisiae*. Here the aptazyme was inserted into the 3′-UTR of the *GAL4* gene. The inactivating mutation is highlighted in yellow, the cleavage site is indicated with a grey arrow. (**g**) Levels of the reporter gene expression (*LacZ*), defined as chemoluminescence divided by the relative $OD_{600}$, of the selected clones in the absence ( − ) and in the presence ( + ) of 100 µg ml$^{-1}$ neomycin in the culture medium. The levels of reporter gene expression in the different conditions are represented by grey tone bars for the controls and red tone bars for the off-switches. The error bars represent s.d. calculated on independent biological triplicates.

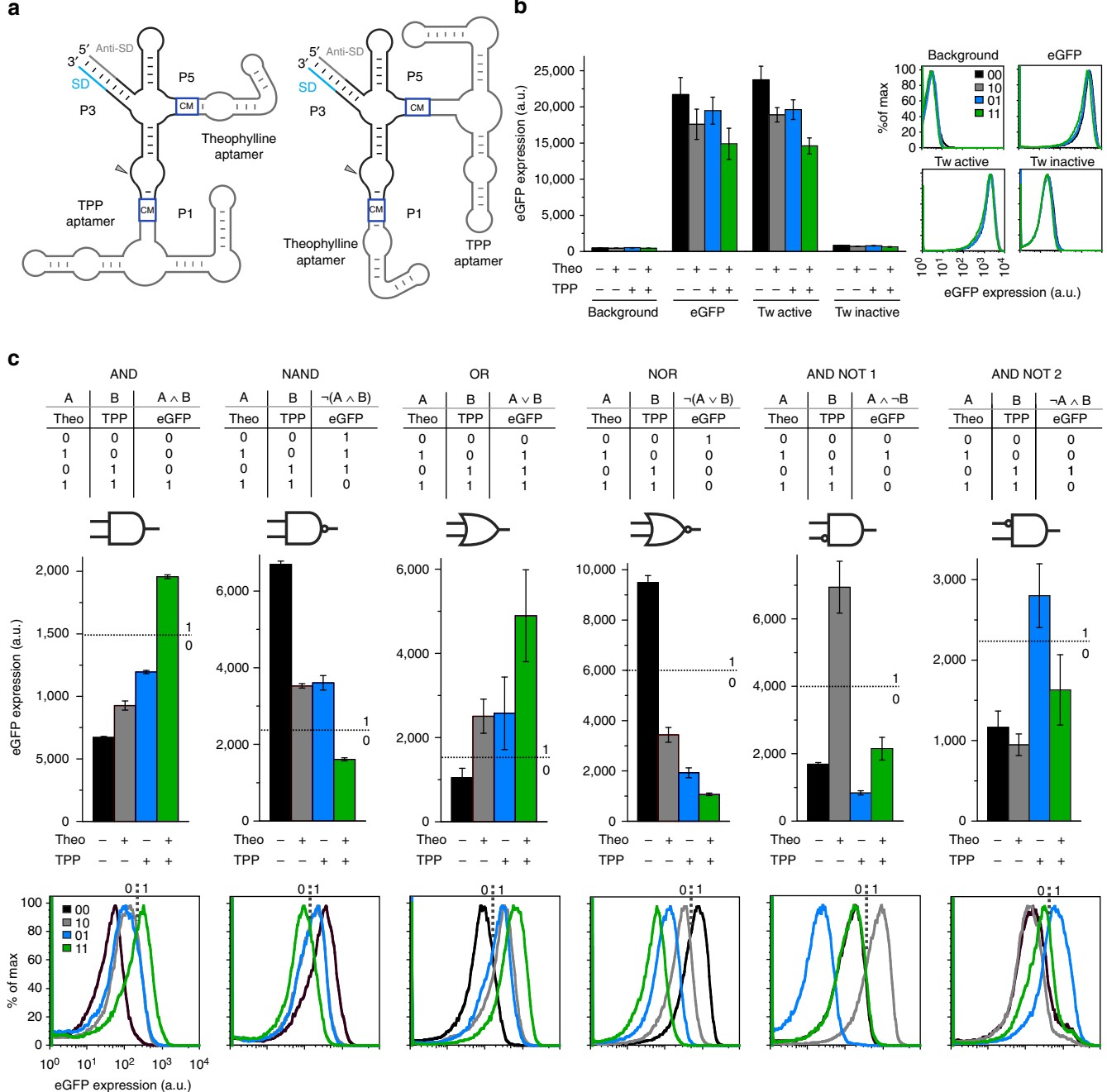

**Figure 3 | Performances of selected twister-based two-input Boolean logic gates.** (**a**) Schematic representation of the twister-based two-input riboswitches. The communicarion modules (CM) are represented with blue boxes. The Shine-Dalgarno (SD) and the anti-SD are shown in light blue and grey respectively. (**b**) Level of eGFP expression (bulk fluorescence divided by the relative $OD_{600}$) and flow cytometry (FC) histograms of the positive and negative controls in the four growth conditions. (**c**) Level of eGFP expression (bulk fluorescence divided by the relative $OD_{600}$) and FC histograms of the Boolean logic gates AND, NAND, OR, NOR, theo_ANDNOT_TPP (ANDNOT1), TPP_ANDNOT_theo (ANDNOT2) are shown. The FC traces of (**b**) the controls and (**c**) the two-input switches were recorded in the absence of ligand (black traces − 0 0), in the presence of 2.5 mM theophylline (grey traces − 1 0), 1 mM thiamin (blue traces − 0 1) and 2.5 mM theophylline/1 mM thiamin (green traces − 1 1). The threshold values, which are represented with dashed lines, allow defining high (value 1) and low (value 0) eGFP expression. The error bars represent s.d. calculated on independent biological triplicates.

exchanging the eGFP gene with the bacterial luciferase luxAB (Supplementary Fig. 22 and Supplementary Note 6).

Finally, five two-input riboswitches were generated using a novel design in which a TPP aptamer is connected to the theophylline aptamer in an 'in-series' fashion. Two libraries were generated attaching the TPP aptamer to two previously developed theophylline P1 switches (off and on respectively) via a communication module of four randomized nucleotides. Although this design seems to be not particularly flexible in terms of the variety of observed switching behaviours, the new constructs show higher absolute levels of eGFP expression (Supplementary Figs 23–25 and Supplementary Note 7). The full sequences of all the two-input twister-based riboswitches are reported in the Supplementary Table 6.

## Discussion

In total we have identified and characterized over 40 different one- and two-input switches differing in input-dependent output logics and switching performances. Such a high diversity of genetic reactivity using a single compact RNA scaffold has never been described before. Interestingly this design is mimicking the situation observed in some naturally occurring twister ribozymes where multiple optional domains are directly connected to the catalytic core. However, whether some of these additional sequences act as ligand-sensing domains has not been investigated so far.

In conclusion, our data demonstrate that the twister motif is a highly versatile RNA scaffold for controlling gene expression in a ligand-dependent manner. Twister ribozymes are flexible expression platforms for artificial riboswitches for a variety of reasons. Twister ribozymes are widespread in both Eukarya and Bacteria[2], increasing the chance for availability of ribozymes from organisms of interest or at least closely related ones. We demonstrated that a twister motif can be engineered to control gene expression in a ligand-dependent way in both a prokaryotic and a eukaryotic model organism. The twister scaffold presents multiple attachment sites (P1, P3 and P5) which can be utilized in order to connect the catalytic domain to the mRNA or to aptameric sensor domains. In the HHR motif the necessity of stems I/II interactions severely limits the re-engineering of ribozymes by incorporating ligand-sensing domains. Hence the compact twister ribozymes should be even more suited for developing complex genetic switches. Interestingly, in some cases of two-input switches even a single nucleotide change in one of the communication modules was sufficient to result in completely different output behaviours (Supplementary Figs 11 and 13). Hence very subtle sequence changes can have great impact on the reactivity of the genetic switch in response to a small molecular ligand, arguing against rational design as a straightforward strategy for generating more complex RNA switches.

In addition to the advances in constructing synthetic RNA-based switches, our results represent the first proof of an engineered twister ribozyme motif capable of regulating gene expression. Although the demonstrated constructs are artificial ligand-dependent versions of a natural motif, our work provides potential roles for these widespread ribozyme motifs in natural contexts. While in our experimental set-up, the cleavage of the 5′-UTR of the mRNA does not seem to influence the stability of the mRNA and the protein expression levels in *E. coli*, in yeast the cleavage of the 3′-UTR by twister strongly affects gene expression. Artificial aptazyme-based riboswitches located in the 3′-UTR of a yeast target gene have been described before[21,29–31]. The first ribozyme-based switches in yeast developed by Smolke and coworkers[21,29] were shown to have switching performances around 2–2.5-fold[33]. The rationally designed HHR-based tetracycline riboswitches of Suess and co-workers showed switching performances up to 2.5-fold[30]. In these works the switching performances were quantified via direct detection of the gene product (a fluorescent protein). In 2014 our group published an *in vivo* selection method for the generation of different HHR-based neomycin riboswitches in yeast, with switching performances up to 25-fold[31]. However, the switching performances of the *in vivo* selected riboswitches were quantified using the same GAL4-LacZ system as described in the present work (see Methods section). For this reason a direct comparison of the switching performances of our neomycin switches to the ones previously published is not possible. We cannot exclude the possibility that our indirect approach of controlling reporter gene expression via control of the expression of a transcription factor possesses an amplification effect. In this case, the reported ribozyme-based switches would work less

efficiently when employed in a direct set-up (that is, inserted immediately into the 3′-UTR of the reporter gene).

Taken together, our data demonstrate that additional structural features in the P1 and P5 sites can be used for modulating twister ribozyme activity, hinting at the possibility of ligand-dependent twister ribozymes existing in nature, as already speculated by Breaker and co-workers[2]. Similarly to our artificial systems, additional structures in naturally occurring ribozymes could be sites enabling the modulation of ribozyme activity and hence gene expression in response to changing conditions.

## Methods

**Expression systems and plasmid construction.** For our study in *E. coli* we employed a plasmid pET16b (Novagen) in which the eGFP gene was inserted under the control of the *T7lac* promoter. The plasmid carries the origin of replication of pBR322 which ensures a number of copies of plasmid per cell around 40 (medium-copy number origin of replication).

The *T7lac* promoter contains a lac operator sequence downstream to the T7 promoter. Although a functional *lacI* gene is present in the genome of *E. coli* BL21(DE3) and a second copy is present in the plasmid, a basal expression of the target protein is present, especially in culture grown to the stationary phase (Novagen pET System Manual). It is important to mention that all our artificial riboswitches were developed taking advantage of the intrinsic leakiness of the system, in the absence of the inducer (IPTG) and in the stationary phase. To further characterize our expression system we sequenced the *lacI* gene present on the plasmid.

We detected a deletion of a single nucleotide (G) at position 1,056. This mutation results in a frame-shift starting from the amino acid 353 and probably in the destruction of the C-terminal helix which forms the tetramerization domain of the Lac repressor[34]. However, we cannot affirm that this mutation is responsible of an increased leakiness of our expression system. In fact at the level of the *T7lac* promoter on the plasmid and of the *lacUV5* promoter, which control the expression of the T7 polymerase gene in the bacterial genome, the repression mechanism does not involve DNA looping and does not require tetramerization of LacI. All the ribozyme and riboswitch constructs presented in the paper were realized inserting an engineered twister ribozyme derived from the env-9 motif (ref. 2) in the 5′-UTR of the eGFP gene. The sequence of the 5′-UTR of these constructs is reported in the Supplementary Fig. 1a–c. The twister ribozyme and the twister-based theophylline and TPP aptazymes developed in *E. coli* were introduced into the pET16b_eGFP plasmid by whole-plasmid PCR with Phusion Hot Start 2 Polymerase (NEB) using primers with the designed ribozyme sequences contained into the 5′-overhang ends (Supplementary Table 7). Subsequently to the PCR, the template plasmid was digested using the enzyme DpnI (NEB). The PCR products were purified by band purification after gel electrophoresis (0.8% GTQ agarose—Roth). The purified products were blunt end ligated (Quick Ligase, NEB) and afterwards transformed into *E. coli* BL21(DE3) gold (Stratagene) by electroporation. The transformed bacteria were plated on Luria–Bertani (LB) agar petri dishes supplemented with 100 µg ml$^{-1}$ carbenicillin and grown aerobically at 37 °C overnight. The catalytically inactive variants of the ribozyme and of the selected twister-based riboswitches were prepared performing site direct mutagenesis by PCR using the suitable pET16b_eGFP containing the aptazyme active form as template. The cloning procedure was performed as described above.

The *luxA* and *luxB* genes were inserted into the pET16b vector containing the selected artificial riboswitches using a traditional cloning approach. The eGFP gene was removed from the pET16b backbone upon digest with NdeI (NEB) and BamHI (NEB) restriction enzymes. The vector was dephosphorylated with Antarctic Phosphatase (NEB) and purified by band purification after electrophoresis (0.8% GTQ agarose—Roth). The *luxA* and *luxB* genes were amplified by PCR from a pUC19-luxAB plasmid using specific primers that introduced the restriction sites for NdeI and BamHI. The insert was digested with the restriction enzymes and purified on column. The purified products were ligated (Quick Ligase, NEB) using a ratio backbone to insert of 1:5 and afterwards transformed into *E. coli* BL21(DE3) gold by electroporation. The transformed bacteria were plated on LB agar petri dishes supplemented with 100 µg ml$^{-1}$ carbenicillin and grown aerobically at 37 °C overnight.

The yeast riboswitches were developed using an engineered form of the pBT3 plasmid, a yeast two hybrid assay (Y2H) plasmid that, using a split-ubiquitin system, is normally employed in membrane protein–protein interactions studies. The open reading frame (ORF) of the fusion protein composed by the C-terminal half of ubiquitin (Cub) and the artificial transcription factor LexA-VP16, was removed by restriction digest and the ORF of the GAL4 transcription factor was inserted on the plasmid (under the control of the *CYC1* promoter). GAL4 activates the transcription of a genomically encoded *lacZ* gene whose expression levels can be easily quantified in a chemoluminescence assay. This plasmid system was already used in our group to develop an *in vivo* selection approach for the isolation of artificial neomycin switches in yeast from a combinatorial library[31]. In this work, *lacZ* expression is used as readout to perform an *in vivo* screening. The long spacer sequence is the 3′-UTR between the riboswitch and the terminator is what remains

on the plasmid after insertion of the GAL4 transcription factor coding sequence. It does not contain any functional RNA sequence or ORF. The sequence of the 3'-UTR of the *GAL4* gene contained in the plasmid pBT3-GAL4 used to develop the neomycin switches in yeast, is reported in the Supplementary Fig. 1d–e. For the construction of neomycin-dependent switches in yeast the twister ribozyme was first inserted into a shuttle vector using a traditional insert cloning approach and the SpeI (NEB) and NotI (NEB) restriction sites. The aptazyme library was created by whole-plasmid PCR as described above and transformed into *E. coli* XL10 gold (Stratagene) by electroporation. After overnight incubation (37 °C, aerobically) on LB agar plates supplemented with 100 μg ml$^{-1}$ carbenicillin the cells were scraped off the plates and incubated overnight in liquid culture (5 ml, 37 °C, 200 r.p.m. shaking). Plasmid DNA was isolated (Miniprep Kit, Quiagen) and digested with SpeI (NEB) and NotI (NEB) restriction endonucleases. The insert was ligated into the pBT3 plasmid containing the same restriction sites.

To confirm successful cloning, the cloned plasmids were isolated (Miniprep Kit, Qiagen) and sequenced (GATC Biotech).

**In vivo screening in *Escherichia coli*.** For the screening, single clones of the transformed pool were picked into 96-deep-well plates and let outgrown to stationary phase in 400 μl of LB medium supplemented with 100 μg ml$^{-1}$ carbenicillin at 37 °C. 10 μl of bacterial culture were re-inoculated in fresh medium in the presence and in the absence of the specific ligands. The bacterial cultures were grown to stationary phase (19 h) shaking vigorously at 37 °C. The screening of the theophylline-dependent switch libraries were performed in LB medium supplemented with 100 μg ml$^{-1}$ carbenicillin in absence and in the presence of 2.5 mM theophylline (Fluka). The screenings of the TPP-dependent switch libraries were performed in M9 medium supplemented with 0.4% glucose and 100 μg ml$^{-1}$ carbenicillin. The final concentration of thiamin (Fluka) was 1 mM. The two-input switch libraries were screened in M9 medium supplemented with 0.4% glucose and 100 μg ml$^{-1}$ carbenicillin. Each clone was grown overnight under four different conditions (No ligand, 2.5 mM theophylline, 1 mM thiamin, 2.5 mM theophylline and 1 mM thiamin).

**Bulk fluorescence measurements.** The eGFP expression levels were evaluated in the different growth conditions performing bulk fluorescence measurements. 100 μl of each culture were transferred into 96-well-microplates and the OD$_{600}$ and the fluorescence of the expressed eGFP (excitation wavelength = 488 nm, emission wavelength = 535 nm) were measured with a TECAN infinite M200 plate reader. The background value was determined measuring the fluorescence of an equally treated *E. coli* BL21(DE3) gold culture carrying a pET16b vector containing a truncated form of the eGFP gene. Fluorescence values were normalized using the respective OD$_{600}$ values.

After the screening, a validation was performed measuring the eGFP expression levels of the selected one– and two-input riboswitches. The bulk fluorescence measurements were performed using a detector gain of 62 if the cells were grown in LB medium or 65 if they were grown in M9 medium. The error bars in the graphs represent the s.d. calculated on independent biological triplicates.

**In vivo screening in *Saccharomyces cerevisiae*.** For the transformation of yeast a heat shock protocol was applied. A colony of *S. cerevisiae* MaV203 was grown overnight in YPAD medium at 30 °C at 200 r.p.m. The culture (2 ml) was centrifuged and washed with 1 ml of sterile water. The cells were centrifuged and resuspended in 50 μl carrier DNA. Plasmid DNA (1 μg) was added together with 240 μl PEG (50%) and 36 μl 1 M lithium acetate. The cells were incubated at 30 °C, 650 r.p.m. for 30 min and heat-shocked at 45 °C for 20 min. The transformed cells were then centrifuged and resuspended in 250 μl of SC-leu medium. 125 μl were plated on SC-leu Petri dishes.

Single clones of the transformed pool were picked and grown in 400 μl of SC-leu medium in 96-deep-well plates at 30 °C shaking vigorously overnight. The next day 10 μl of each yeast culture were transferred to a new 96-deep-well plate containing 400 μl SC-leu medium in the presence and in the absence of neomycin (100 μg ml$^{-1}$). The plates were incubated overnight (30 °C, 1350 r.p.m.). The expression of *lacZ* reporter gene was measured using the GalScreen assay (Applied Biosystems). The assay was performed according to the supplied protocol using 25 μl of yeast culture per well. For each culture 100 μl were used for OD$_{600}$ measurement. Luminescence was measured in black 96-well half area plates using a Tecan infinite M200 plate reader (integration time 2000 ms). Luminescence data were normalized to the OD$_{600}$ values and the background was subtracted. DNA was isolated from the clone of interest lysing liquid yeast cultures in a FastPrep-24 instrument. Plasmids were isolated afterwards using a Zyppy Plasmid Miniprep Kit. To obtain sufficient concentrations for sequencing, the plasmids were transformed into *E. coli* XL10, enriched, purified and prepared for sequencing (GATC biotech).

**Reverse transcription semiquantitative real-time PCR analysis.** Total RNA was isolated from *E. coli* using the RNeasy Mini Kit (QIAGEN) with additional RNA protect Bacteria Reagent (QIAGEN) and doubled DNAse I digest and phenol extraction. Total RNA was purified from 5 ml bacterial cultures grown to the late exponential phase (OD$_{600}$ = 0.8) in LB medium supplemented with 100 μg ml$^{-1}$

carbenicillin at 37 °C. The cultures were inoculated from an overnight culture to OD$_{600}$ of 0.1. Cells ($2.5 \times 10^8$) were harvested and pelleted by centrifugation for 10 min at 5,000*g*. Cell pellets were lysed by addition of 1 mg ml$^{-1}$ proteinase K and 25 mg ml$^{-1}$ lysozyme (Roth) in 1 × TE buffer, resuspended and incubated 10 min at room temperature. RNA was purified according to the QIAGEN RNeasy Mini Kit manual. Genomic DNA was removed by doubled DNAse I digest (3 U per run) for 20 min at 37 °C. Phenol extraction was performed with Roti-Aqua phenol pH 4.5–5.0 (Roth) followed by 100% chloroform extraction. RNA was precipitated by 1/10 volumes of 3 M sodium acetate buffer pH 5.2 and 3 volumes of 100% ethanol at − 20 °C overnight. The purified pellet was dissolved in ultrapure water. RNA concentration was determined using Tecan infinite M200 equipped with Nano Quant Plate (software Tecan i-comtrol). Purity of samples was given by A$_{260}$/A$_{280}$ ratio of > 2.0. RNA was separated on 0.8% agarose gel electrophoresis and the bands were visualized and quantified after peqGreen staining in Amersham Imager 600. The ratios of 23 S rRNA to 16 S rRNA were ∼1.5–2.0, pointing towards good RNA integrity. Synthesis of complementary DNA (cDNA) and non-reverse transcribed controls were performed using 1 μg total RNA and a random hexamer primer in 20 μl reaction volume with Superscript III Reverse Transcriptase (SIIIRT 200 U, Invitrogen). Digest with 5 U RNaseH for 20 min at 37 °C and following heat inactivation at 65 °C, 20 min removed RNA from RNA to DNA heteroduplexes in the last step. For the detection of the target gene (eGFP) the following primers were used (target length 164 nt):

F primer: 5′-GAAGGAGATATACCATGGGCCATCA-3′
R primer: 5′-GCTGAACTTGTGGCCGTTTAC-3′
For the reference gene *ssrA* (target length 65 nt):
F primer: 5′-ACGGGGATCAAGAGAGGTCAAAC-3′
R primer: 5′-CGGACGGACACGCCACTAAC-3′.

qPCR was performed on a TOptical Thermocycler (Biometra, Analytic Jena) in 96-well plates in a final volume of 10 μl (1 × HF buffer, 200 μM dNTPs, 600 nM each primer, 0.04 μl cDNA from synthesis volume, 3 × SYBR Green I (Sigma) in DMSO, 3 U Phusion Hot Start II polymerase (Biozym) and 5.36 μl DEPC treated steril water). The enzyme mixture was added together with the cDNA in 6 μl to 96 well, a primer mix was added in a total volume of 4 μl to the 96 well, performed in single manual pipetting steps. Following thermocycler parameters have been used: initial denaturation at 98 °C, 30 s; denaturation 98 °C, 10 s, annealing 65 °C, 30 s and extension 72 °C; 20 s in 45 cycles, followed by a final extension of 72 °C, 7 min. Specificity of qPCR was determined by melting curve analysis performed after the final extension, starting from 65 °C up to 98 °C. Melting curves resulted in single product specific melting temperatures of *ssrA*, 87.6 °C and eGFP, 91.3 °C. C$_t$ values were determined using the qPCR 3.1 software (Biometra, Analytik Jena).

**Measurements of *in vitro* aptazyme activity.** Synthetic DNA templates of four different theophylline- and TPP-dependent aptazymes were generated by Taq-PCR (NEB) using the following primers:

R primer: 5′-CTCCTTTAAAGCTGCATTAA-3′;
F primer (active forms): 5′-GAATTAATACGACTCACTATAGGGAGCTCC TTTAAAGCGGTTACAAGCC-3′;
F primer (inactive forms): 5′-GAATTAATACGACTCACTATAGGGAG CT CCTTTAAAGCGTGTACAAGCC-3′.

The total PCR reaction volume was 200 μl. The PCR products were ethanol precipitated (1/10 volume of 3 M sodium acetate pH 5.7 and 3 volumes of 100% ethanol—Sigma-Aldrich). The pellets were resuspended in 40 μl of ddH$_2$O. The PCR products were *in vitro* transcribed using T7 RNA polymerase (Thermo Scientific) in Thermo Scientific transcription buffer, 90 nM ATP, 2 mM CTP, 2 mM GTP and 2 mM UTP, 80 U RiboLock RNase Inhibitor (Thermo Scientific), 0.15 U of PPase (Thermo Scientific), 5 μCi $^{32}$P-α-ATP and in presence of 25 μM blocking strand (5′-TACTCTGCTATTTTTGCGGGCTTGTA-3′). This last measure prevents self-cleavage of *in cis* ribozymes during transcription. After 2 h of incubation at 37 °C the reactions were ethanol precipitated overnight and resuspended in 40 μl of ddH$_2$O. One volume of 2 × loading buffer (80% (v/v) formamide, 50 mM EDTA pH = 8.0) was added. Subsequently the products were purified by 8% denaturing PAGE. Full-length products were excised and extracted from gel using 500 μl Rotipuran water (Roth) supplemented with 1 mM EDTA. All the following steps were performed using Rotipuran water to ensure the absence of Mg$^{2+}$.

Activities were determined in 50 mM Tris-HCl pH 7.5, 0.1 M KCl in the presence and in the absence of the respective ligand (2.5 mM theophylline or 1 mM TPP—Fluka). The final concentration of the RNA in the reaction mix was 50 nM.

The folding was performed incubating the reaction mix at 95 °C for 2 min and then slowly cooling down to 37 °C. Cleavage reaction was started by addition of Mg$^{2+}$ to a final concentration of 1 mM. Reactions were quenched with loading/ stop buffer (80% (v/v) formamide, 50 mM EDTA pH = 8.0, 0.025% (w/v) bromophenol blue and 0.025% (w/v) xylene cyanole) after defined time points. The reaction mix was separated by 8% PAGE. The results were visualized by phosphorimaging and the bands were quantified.

The kinetic traces were fitted using two different equations:

$$y = A \cdot e^{-x/t} + y_0 \tag{1}$$

$$y = A_1 \cdot e^{-x/t_1} + A_2 \cdot e^{-x/t_2} + y_0 \tag{2}$$

where *y* is the fraction cleaved at the time *x*.

**Flow cytometry analysis.** For flow cytometry analysis bacterial cultures were grown as described for the bulk fluorescence measurements in 96-deep-well plates in the presence and the absence of the respective ligands. 400 μl of culture were harvested, pelleted and washed once in $1 \times$ PBS. The pellet was resuspended in 1.5 ml $1 \times$ PBS and analysed with a FACScalibur cell analyser (BD Biosciences Singapore, FlowKon facility) using a 488 nm argon laser for excitation. Fluorescence of 100,000 cell counts was detected through FL-1 band pass filter with photon multiplier tube voltage of 840. Additionally, forward scatter and side scatter were measured. Logarithmic signal amplification was used. All cytometric data were gated as shown in the Supplementary Fig. 6 and in the Supplementary Tables 1 and 5 and analysed with FlowJo software. Populations were compared in histograms with counts plotted against FL-1 signal. The T(χ) metrics was calculated for the different couples of populations compared. T(χ) is a statistic parameter that not only provides an indication of the probability with which two distributions are different, but also it allows to rank the difference between different samples[4]. Only couples of populations which have T(χ) values larger than an empirically determined minimum can be considered to be different.

**Bioluminescence assay.** For the bioluminescence assay, bacterial cultures were grown as described for the bulk fluorescence measurements in 96-deep-well plates, in M9 medium in the presence and the absence of the respective ligands. 400 μl of culture were harvested, pelleted and washed once in $1 \times$ PBS. The pellets were resuspended in 600 μl $1 \times$ PBS. 100 μl from each culture were transferred into 96-well-microplates and the $OD_{600}$ was determined. Chemoluminescence was measured with a TECAN infinite M200 plate reader equipped with an automatic injection system and using full-area flat black 96-well-microplates. Luminescence data were acquired every three seconds for a total of one minute after the addition of 10 μl of 0.01% decanal in ethanol solution to 100 μl of bacterial culture and using an integration time of 2000 ms. The background luminescence was quantified using an inactive LuxAB clone carrying the H44A mutation in the *luxA* gene. The data were normalized using the respective $OD_{600}$ values end expressed as relative chemoluminescence by dividing for the maximum luminescence value recorded for each biological replicate.

**Data availability.** The authors declare that all data supporting the findings of this study are available within the article and its Supplementary Information Files and from the authors upon reasonable request.

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

## Acknowledgements

We thank Dr Stefanie Bürger and the flow cytometry facility of the University of Konstanz (FlowKon) for technical support, Dr Christiaan Karreman for providing the plasmid pUC19-luxAB encoding the bacterial luciferase reporter, Dr Benedikt Klauser and Johann Bauer for helpful suggestions and scientific discussions. Financial support from the Deutsche Forschungsgemeinschaft (DFG)-funded initiative CRC 969 'Chemical and Biological Principles of Proteostasis' is acknowledged.

## Author contributions

J.S.H. conceived this project; M.F. and J.S.H. designed the experiments; M.F., J.S., L.A.W. and S.G. performed the experiments and analysed the data, M.F. and J.S.H. prepared the manuscript.

## Additional information

**Competing financial interests:** The authors declare no competing financial interests.

