## [Peer review file · Nature Communications]

NCOMMS-16-03504C

Title:

Twister ribozymes as highly versatile expression platforms for artificial riboswitches

PEER REVIEW FILE

From NCOMMS-16-03504-T

Reviewer #1 (Remarks to the Author):

This revised manuscript by Felletti et al. has nicely addressed most of my comments made on the initial draft. I have the following specific comments on the revised manuscript and the author's responses:

The main part of Comment #1 really could not be addressed by the authors, but regardless it still would be good to see this work published in a high-profile journal such as Nature Communications. I do hope this work is published without additional delay.

Comments #2 through #6 have been satisfactorily addressed.

The author's response to Comment #7 is not what I had expected, although I probably worded my concern in a fashion that was too vague. To be clearer, I do not believe that twister ribozymes are 100- to 500-fold faster than other self-cleaving ribozymes. The actual speeds for twister ribozymes have not been measured to be faster than hammerhead ribozymes, or even that some other self-cleaving ribozymes. Rather, the projected rate constants for twister are as much as 1000 per minute, but the actual or projected rates of most other ribozymes will be similar. The way to easily resolve this issue is to delete the comments implying that twister is a superior component of engineered RNAs because of its superior speed.

Comment #8 has been satisfactorily addressed.

Reviewer #2 (Remarks to the Author):

The authors have partially addressed some of the concerns but there are still some issues:

(1) To address the concern about the 2-input riboswitches responding to both inputs in the same cell, the authors have presented flow cytometry data that does not directly address the concern.

Furthermore, since there are wide overlaps in many flow cytometry histogram traces (Sup.Fig.19b), this data raises serious doubts about the claimed logic-gate behaviours. To convince anyone of these data, the authors must present p-values to demonstrate for each arbitrary threshold that each fluorescence trace below it is significantly different from each trace above it. For example, in the NAND gate example (TheoTPP 1.4) the fluorescence trace of the 00-input (below threshold) should be significantly different when compared to each of the 10-input, 01-input, and the 11-input (above threshold).

(2) They have not performed the SHAPE analysis, but have argued that using catalytically inactive mutants serves the same purpose.

(3) They have not quantified bands on the RNA gels to measure cleavage percentage, and have explained that the intended purpose of the experiment was to demonstrate "qualitatively" that the catalytically inactive mutants indeed behave as claimed. Effectively, they are choosing to ignore the obvious observation in the same gel that the effect of small-molecule addition on cleavage is rather limited.

(4) They have added some more discussion, but there is still huge scope for improvement. The

authors may want to explain their various results (why P3-SD design works better than P1-SD design, why P1-aptamer design works better than P5-aptamer) based on biophysical principles (Borujeni et al., 2013 & 2015).

(5) They have not analysed their riboswitches in mid-log phase of growth, and have argued that since riboswitches are regulated at the level of translation, stationary phase is the ideal phase of growth in which to characterise them. They cite Gefen et al., 2014 to support this argument. However, this seems to be a mis-reading of that work where: (1) genes are induced for activation after cells have entered the stationary phase unlike in the present study, (2) gene activation being studied is in fact transcriptional activation by inducible promoters and not translation activation, and (3) cells being studied are being maintained in a controlled microfluidics environment during starvation unlike the batch culture used in this study. Using mid-log phase expression data would allow confirmation that expression levels have reached steady state, compare growth rates, as well as avoid any secondary effects of the inducer.

(6) Although they have used a different reporter (luciferase) to test their 2-input riboswitches, the cells used are still those in stationary phase of growth.

Reviewer #3 (Remarks to the Author):

In this manuscript, Hartig and co-workers show that the twister ribozyme can serve as a flexible expression platform in synthetic riboswitch designs. In particular, they demonstrate that the twister ribozyme platform can be used to generate many one and two-input regulators by connecting two different aptamers (theophylline and TPP) at two different sites. They use the two-input ribozyme platform and screen for variants exhibiting a wide variety of gene regulatory behaviors which can be represented using binary Boolean logic gates including AND, NAND, OR, NOR, and ANDNOT operators in bacteria. Finally, they show that the twister ribozyme platform can be coupled to the neomycin aptamer and used to control gene expression in yeast.

This is novel work in the sense that this is the first time the twister ribozyme has been used to generate ligand-responsive genetic riboswitches. The writing is clear and the data is technically sound. Importantly, the main claim that "the twister ribozyme is distinguished as an outstandingly flexible expression platform" is well supported by the work. Specifically, the data presented in the main text and in the supplemental provide sufficient evidence of this flexibility - mainly that:

- different aptamers (natural and synthetic) can be incorporated into the twister ribozyme
- two different sites for incorporating aptamers are well-supported
- the ribozyme can support two-inputs and be used to represent a variety of gene regulator behaviors (e.g., Boolean logic gates AND, NAND, OR, NOT, ANDNOT)
- the ribozyme can be used for control in both bacteria and yeast

This work will be of interest to RNA synthetic biologists as the ribozyme may be a useful addition to the genetic engineering toolkit.

Other more specific points that should be addressed prior to publication:

1. As previously noted, much of the field has moved to characterization through flow cytometry. The authors have now characterized their main riboswitches by flow cytometry and included this data in the supporting information. It is recommended that the flow cytometry be directly used to demonstrate the gene expression changes in the main text figures (Fig 2 and Fig 3) rather than the

bulk fluorescence. The authors should also clearly describe in the methods how their flow cytometry data was collected, processed, and analyzed as different strategies are used in the field.

2. There is very little information about the generation of the neomycin riboswitch. Was only one riboswitch found? Why is only an off-switch reported? Is this a limitation of the ribozyme platform? Why is there a long spacer sequence between the riboswitch and the terminator sequence? Why was the P1 site chosen for integration of this aptamer? Did P5 not work? These types of design choices are important to discuss for the broader impact of this work.

3. The choice of characterizing the neomycin riboswitch using the Gal4 gene is strange and inconsistent with the rest of the manuscript, i.e., all other riboswitches in this work were characterized using GFP. It would be easier to interpret the activity of this riboswitch relative to the others reported in this manuscript and in from other work in the field if it were characterized regulating the expression of GFP. How was the Gal4 expression measured (this is not in the experimental methods)? Also the error bars for the neomycin riboswitch in Figure 2e are very large (which may be related to the assay method). The authors may want to characterize this riboswitch in a way that is consistent with their other assays to allow for more direct comparisons; as it is presented, it is hard to draw conclusions regarding the performance of this particular OFF switch.

Response to editorial and reviewer comments:

Editorial comments:

After discussion your manuscript with the editorial team, we request a revision of the sentence "Although Smolke and co-workers reported HHR-based two-input Boolean operators in yeast, the reported performance of these switches has been questioned due to the application of unusual normalization procedures¹⁸." It is our understanding that the work cited does not question the normalisation procedures in the work by Smolke et al. Please provide additional references supporting this controversy or revise the sentence to reflect the work cited.

Response: The sentence that was called into question was rephrased and a reference was added. According to the cited reference (Chen and Ellington 2009 PLOS computational biology):

"The discrepancy between the interpretation and the data was due to redefinition of the word 'fold' by the authors. Although the word 'fold' is generally used to express the ratio of two quantities, Win and Smolke used 'fold' as a unit of absolute quantity of GFP expression. For example, the GFP expression level from an unengineered plasmid was defined as '50 fold.' Therefore, when the GFP expression level from an engineered plasmid changed from '20 fold' in the absence of theophylline to 43 fold' in the presence of theophylline, a dynamic range of '(43-20=) 23 fold' could be claimed. Most researchers would instead estimate the dynamic range to be (43/20 =) 2.2-fold. Win and Smolke have also reported that multiple aptazymes inserted into the 3'-UTR could act as logic gates for gene expression, but the raw data necessary to evaluate these claims were not immediately available."

We therefore invite you to revise and resubmit your manuscript, taking into account the points raised. At the same time, we ask that you ensure your manuscript complies with our editorial policies.

The editorial policies were taken into account, see the uploaded checklists.

Reviewers' comments:

Reviewer #1 (Remarks to the Author):

This revised manuscript by Felletti et al. has nicely addressed most of my comments made on the initial draft. I have the following specific comments on the revised manuscript and the author's responses:

The main part of Comment #1 really could not be addresses by the authors, but regardless it still would be good to see this work published in a high-profile journal such as Nature Communications.

I do hope this work is published without additional delay.

Comments #2 through #6 have been satisfactorily addressed.

The author's response to Comment #7 is not what I had expected, although I probably worded my concern in a fashion that was too vague. To be clearer, I do not believe that twister ribozymes are 100- to 500-fold faster than other self-cleaving ribozymes. The actual speeds for twister ribozymes have not been measured to be faster than hammerhead ribozymes, or even that some other self-cleaving ribozymes. Rather, the projected rate constants for twister are as much as 1000 per minute, but the actual or projected rates of most other ribozymes will be similar. The way to easily resolve this issue is to delete the comments implying that twister is a superior component of engineered RNAs because of its superior speed.

Comment #8 has been satisfactorily addressed.

Response: The sentence was removed according to reviewer's suggestion.

Reviewer #2 (Remarks to the Author):

The authors have partially addressed some of the concerns but there are still some issues:

(1) To address the concern about the 2-input riboswitches responding to both inputs in the same cell, the authors have presented flow cytometry data that does not directly address the concern. Furthermore, since there are wide overlaps in many flow cytometry histogram traces (Sup.Fig.19b), this data raises serious doubts about the claimed logic-gate behaviours. To convince anyone of these data, the authors must present p-values to demonstrate for each arbitrary threshold that each fluorescence trace below it is significantly different from each trace above it. For example, in the NAND gate example (TheoTPP 1.4) the fluorescence trace of the 00-input (below threshold) should be significantly different when compared to each of the 10-input, 01-input, and the 11-input (above threshold).

In the comments to the first version of the manuscript the reviewer required to show that our switch "work at the single cell level". In the artificial riboswitch field this is normally done by characterizing the switches by flow cytometry (FC) (Win et al PNAS 2007, Win et al Science 2008, Lynch et al NAR 2009 are just some examples). Our riboswitches are now fully characterized using three different methods (fluorescence bulk measurements, flow cytometry, chemiluminescence). The description at the level of the "same single cell" is actually a technically very demanding task and the request is very unusual since it is not common practice in gene expression studies. Moreover the measurements at the level of the same single cell exposed to the different inputs would require taking in consideration

the protein turnover for the evaluation of the switch performances. More importantly we don't think that such measurements would add fundamental knowledge about our switches that were already shown to work with well-established methods and protocols.

Regarding the statistical analysis of the flow cytometry data, a statistical comparison of the population (histograms) was performed. The software for the analysis of the FC data (FlowJo) utilizes two different algorithms for the comparison of the two FC populations. The Kolmogorov-Smirnoff statistic test (a non parametric analogous of the chi square test) provides a probability that two FC histograms are different. However it is reported to be too sensitive to provide meaningful values. Indeed all our population comparisons were shown to be statistically significantly different, including the controls. We decided to compare our populations using the probability Binning (PB) algorithm present in the FlowJo platform. The latter was shown to be able to detect small differences between histograms, moreover it provides with a $T(X)$ metric allows to rank different samples (Roederer et al. Cytometry 2001). Population comparison was performed for each FC histogram in the presence and in the absence of ligand (or in the case of the two-input switches between each combination of the four culture conditions tested). $T(X)$ values were calculated and reported in the Supplementary Information (Supplementary Figs. 6,8,20,21). According to the FlowJo manual $T(X)$ is a statistic which provides an indication of the probability with which two distributions are different: The higher the value of $T(X)$, the less like the control sample the test sample is. When $T(X) = 0$, the two histograms are indistinguishable ($p = 0.5$). A value $T(X) > 4$ implies that the two distributions are different with a $p < 0.01$ (99% confidence). According to the FlowJo user manual, however, the minimum value of $T(X)$ that has biological significance depends on the nature of the data being analyzed and therefore needs to be determined empirically. Only populations which have $T(X)$ values larger than this empirical minimum can be considered to be different. The $T(X)$ values for the controls (background control, eGFP control, Twister active and twister inactive) were also calculated. All the $T(X)$ values of the one-input and the relevant $T(X)$ values for the definition of the logic gate in two-input switches, were shown to be significantly higher than the higher $T(X)$ value calculated for the four control samples (including the NAND gate TheoTPP 1.4). Taken together, the results of comparing $T(X)$ values demonstrate that the FC histograms recorded in the different input conditions are significantly different. We have added this analysis to the Supporting Information and $T(x)$ values to the Supplementary Figs. 6, 8, 20, 21).

(2) They have not performed the SHAPE analysis, but have argued that using catalytically inactive mutants serves the same purpose.

We would like to specify that we didn't argue that "using catalytically inactive mutants serves the same purpose of a SHAPE analysis". The catalytically inactive negative control presents a very similar sequence to the one of the active motif, but it is cleavage incompetent. The use of catalytically inactive motifs is a standard practice in the field of ribozyme-based artificial gene switches. For our work, the only important requirements are (i) that the sequence of the inactive motif is highly similar to the one of the active and (ii) that it is catalytically inactive (as already shown by Breaker and coworkers). We think that a detailed structural characterization of the active and catalytically inactive switches, although interesting, goes far beyond the scope of the present work. In addition (and this is the most important point why we are convinced that a SHAPE analysis of our ribozymes will not result in meaningful insights), it is not clear what would be the ribozyme species that contributes to the SHAPE signals: Since we have a reacting system, is the pre-cleavage sequence probed or are cleavage fragments contributing to the results? In order to address this very important question one could then use inactivated sequences, however, as the reviewer correctly argues, they very likely have different structures. Hence such an analysis is inconclusive and consequently has to our knowledge never been carried out in this field.

(3) They have not quantified bands on the RNA gels to measure cleavage percentage, and have explained that the intended purpose of the experiment was to demonstrate "qualitatively" that the catalytically inactive mutants indeed behave as claimed. Effectively, they are choosing to ignore the obvious observation in the same gel that the effect of small-molecule addition on cleavage is rather limited.

The figure called into question was removed. The figure was redundant because the demonstration of the efficacy of the inactivating mutation of the twister ribozyme was already provided by Roth et al 2014. Moreover a quantification of the effects of the ligands on the active forms of the switches is already present in the Supplementary Fig. 9 together with an extended discussion of the kinetics data in the Supplementary Note 4.

(4) They have added some more discussion, but there is still huge scope for improvement. The authors may want to explain their various results (why P3-SD design works better than P1-SD

design, why P1-aptamer design works better than P5-aptamer) based on biophysical principles (Borujeni et al., 2013 & 2015).

In the present version of the manuscript we improved:

- The introduction about twister ribozyme, including the findings of some recent works.
- We cite the c-di-GMP-dependent group I intron as an example of naturally occurring ligand-dependent ribozymes.
- We improved the description of the employed yeast expression system (Supplementary Note 1).
- We extended the discussion about our neomycin riboswitches in yeast comparing them with the previously engineered riboswitches in yeast (Supplementary Note 5)
- We improved the discussion of the FC data, including a quantification of the distances between the populations and a description of the gating procedure (Supplementary Figs. 6-8-20-21 and Supplementary Notes 3 and 6)
- We cited the work of Borujeni et al. NAR 2013 to explain why P3-SD design works better than P1-SD design (main text – RESULTS)
- We cited the work of Borujeni et al. NAR 2016 to explain the effect of co-transcriptional folding and molecular crowding on activity of riboswitches at the translational level (Supplementary Note 4)

(5) They have not analysed their riboswitches in mid-log phase of growth, and have argued that since riboswitches are regulated at the level of translation, stationary phase is the ideal phase of growth in which to characterise them. They cite Gefen et al., 2014 to support this argument. However, this seems to be a mis-reading of that work where: (1) genes are induced for activation after cells have entered the stationary phase unlike in the present study, (2) gene activation being studied is in fact transcriptional activation by inducible promoters and not translation activation, and (3) cells being studied are being maintained in a controlled microfluidics environment during starvation unlike the batch culture used in this study. Using mid-log phase expression data would allow confirmation that expression levels have reached steady state, compare growth rates, as well as avoid any secondary effects of the inducer.

(6) Although they have used a different reporter (luciferase) to test their 2-input riboswitches, the cells used are still those in stationary phase of growth.

Our switches were not characterized in the exponential phase. First of all we should mention that already many works were published in the past where the switches were designed, selected and/or characterized in the stationary phase (Wieland et al. 2008, 2009, 2010, Saragliadis et al. 2013, Klauser et al. 2012, 2013, Carothers et al. Science 2011). It's true that many other were

characterized in the exponential phase, but we can say for sure that there is not a common consensus about the phase in which a bacterial artificial riboswitch should be characterized. Reviewer 2 argues that the switches using the mid-log phase allows to check that expression levels have reached steady state, compare growth rates to avoid secondary effects of the inducer. We do not agree on this point. Measuring in the mid-log phase does not ensure that the expression levels have reached steady state and the comparison of the growth rates. Measuring the gene reporter expression at a special OD600 value (different groups have used different OD600 values) provides information exclusively about the gene expression levels at that specific growth phase exactly as we did in the stationary phase. Moreover secondary effects of the inducers can be easily evaluated comparing the gene reporter expression levels of the controls. In this work we show a number of controls (positive control, background control, twister ribozyme constitutive active, twister ribozyme constitutive inactive). For none of them a significant secondary effect was observed neither in the fluorescence measurements (bulk, flow cytometry) nor in the chemoluminescence. This much stronger evidence arguing against secondary effects on gene expression than observing growth curves. In general, the mid-log phase would be more prone to secondary effects of the ligands. In the work of Borujeni et al. 2016 the artificial switches were characterized in the exponential phase, however a direct effect on the gene reporter expression was observed. This effect was evaluated comparing the gene expression level of the positive control in the presence and in the absence of the ligands (exactly as we do!).

Reviewer #3 (Remarks to the Author):

In this manuscript, Hartig and co-workers show that the twister ribozyme can serve as a flexible expression platform in synthetic riboswitch designs. In particular, they demonstrate that the twister ribozyme platform can be used to generate many one and two-input regulators by connecting two different aptamers (theophylline and TPP) at two different sites. They use the two-input ribozyme platform and screen for variants exhibiting a wide variety of gene regulatory behaviors which can be represented using binary Boolean logic gates including AND, NAND, OR, NOR, and ANDNOT operators in bacteria. Finally, they show that the twister ribozyme platform can be coupled to the neomycin aptamer and used to control gene expression in yeast.

This is novel work in the sense that this is the first time the twister ribozyme has been used to generate ligand-responsive genetic riboswitches. The writing is clear and the data is technically sound. Importantly, the main claim that "the twister ribozyme is distinguished as an outstandingly flexible expression platform" is well supported by the work. Specifically, the data presented in the main text and in the supplemental provide sufficient evidence of this flexibility - mainly that:
-different aptamers (natural and synthetic) can be incorporated into the twister ribozyme

- two different sites for incorporating aptamers are well-supported
- the ribozyme can support two-inputs and be used to represent a variety of gene regulator behaviors (e.g., Boolean logic gates AND, NAND, OR, NOT, ANDNOT)
- the ribozyme can be used for control in both bacteria and yeast

This work will be of interest to RNA synthetic biologists as the ribozyme may be a useful addition to the genetic engineering toolkit.

Other more specific points that should be addressed prior to publication:

1. As previously noted, much of the field has moved to characterization through flow cytometry. The authors have now characterized their main riboswitches by flow cytometry and included this data in the supporting information. It is recommended that the flow cytometry be directly used to demonstrate the gene expression changes in the main text figures (Fig 2 and Fig 3) rather than the bulk fluorescence. The authors should also clearly describe in the methods how their flow cytometry data was collected, processed, and analyzed as different strategies are used in the field.

The flow cytometry data were included in Fig 2 and Fig 3 in the main text as requested by referee 3. Additional information about FC measurements and data processing and significance testing were added to the methods and the Supplementary Information (Supplementary Note 3). A new Supplementary Fig 6 and new Supplementary Tables 1 and 5 were added to provide details about the gating procedure. Statistical comparisons of the FC histograms were also performed (Supplementary Figs. 6, 8, 20, 21).

2. There is very little information about the generation of the neomycin riboswitch. Was only one riboswitch found? Why is only an off-switch reported? Is this a limitation of the ribozyme platform? Why is there a long spacer sequence between the riboswitch and the terminator sequence? Why was the P1 site chosen for integration of this aptamer? Did P5 not work? These types of design choices are important to discuss for the broader impact of this work.

An additional figure and further information and discussion were added to the Supplementary Information (Supplementary Note 1 and 4, Supplementary Fig.11). In particular, in the present version of the manuscript we added a further neomycin off-switch. The screening of the neomycin aptamer in P1 we picked the three switches that showed best performances. Following sequencing two of them showed the same sequence. No switch was isolated from the screening in P5. In the Supplementary Note 1 we provide more information about the employed plasmid system in yeast and we explain that the long spacer sequence in the 3'-UTR between the riboswitch and the

terminator is what remains after insertion of the GAL4 transcription factor coding sequence on the plasmid. The spacer does not contain any functional RNA sequence and it does not play a role in the switching activity of the neomycin switch.

3. The choice of characterizing the neomycin riboswitch using the Gal4 gene is strange and inconsistent with the rest of the manuscript, i.e., all other riboswitches in this work were characterized using GFP. It would be easier to interpret the activity of this riboswitch relative to the others reported in this manuscript and in from other work in the field if it were characterized regulating the expression of GFP. How was the Gal4 expression measured (this is not in the experimental methods)? Also the error bars for the neomycin riboswitch in Figure 2e are very large (which may be related to the assay method). The authors may want to characterize this riboswitch in a way that is consistent with their other assays to allow for more direct comparisons; as it is presented, it is hard to draw conclusions regarding the performance of this particular OFF switch.

We would like first of all to thank the reviewer to raise this issue. In the past years ribozyme-based riboswitches in yeast with rather modest switching performances were published. The first ribozyme-based switches in yeast developed by Smolke and coworkers (PNAS 2007 and Science they were shown later to have switching performance around 2 - 2.5 folds (see Chen and Ellington 2009 PLOS computational biology for an accurate calculation of the fold of activation and inactivation). The best rationally designed tetracycline switches of Sues and co-workers were able to repress gene expression up to 2.5-fold. However in the reporter assays of Smolke and Sues, the artificial riboswitch is inserted directly into the 3'-UTR of the reporter gene (a fluorescent protein). In 2014 our group published an in vivo selection method for the generation of HHR-based neomycin riboswitches in yeast (Klauser et al ACS Synthetic Biology 2014). In this work the quantification of the switches performances was done employing the same system as illustrated in the present manuscript. The riboswitch is controlling the expression of the GAL4 transcription factor which in turn is promoting the expression of the beta-galactosidase. We have added further explanation to the Supporting Information in order to address the suggestions of reviewer 3.

Using this system the performances of the switches published in 2014 were up to 25-fold and in this work up to 10-fold. The beta-galactosidase assay was repeated including the previously not included Tw_Neo_2. The new data are now presented in Fig. 2 and in the Supplementary Fig. 11. We decide to use this expression system because it ensures sufficient levels of reporter gene expression to perform the in vivo screening. We were not able to obtain sufficient level of eGFP expression to perform a screening. Following the observation of the reviewer, we inserted active and inactive forms of the twister ribozyme as well as our neomycin switches into the 3'-UTR of an eGFP construct contained in the p413 plasmid (yEGFP as a reporter gene, CYC1 terminator, GPD promoter, BY4741 yeast strain). Unfortunately this resulted in very low expression levels of eGFP,

in a poor differential expression of eGFP when an active or an inactive form of the twister is inserted in the 3'-UTR (less than 2-fold) and in little switching performance when the neomycin switches were inserted. Hence in order to perform a meaningful comparison of the switches, additional optimization of the reporter system would be required. However, direct comparison of the performance of switches from different studies is complicated because the choice of plasmid, promoter, yeast strain, type of reporter eGFP etc. potentially influences the results. In order to address the reviewers' suggestions we have added a critical discussion of the reporter system that explicitly mentions the possibility that the performance of the switches could decrease when employed in a direct setup (Supporting Information, Supplementary Note 5).

Reviewers' comments:

Reviewer #2 (Remarks to the Author):

The authors have addressed most of our remaining comments. However, their findings and results for the third comment are notable. Specifically, their findings clearly show that switches will have different performances when characterized in different reporter systems (i.e., a direct fluorescent readout (GFP) versus an indirect enzymatic reporter readout (B-gal)). These results highlight the point that it is difficult (if not impossible) to make performance comparisons across papers that characterize switches in different assays, systems, etc; and that direct comparisons require that the switches be characterized in the same exact assay in the same experiment.

It's expected that enzymatic assays will be more sensitive and thus provide greater sensitivity to smaller activity readings (and greater reported dynamic ranges) than a direct fluorescent readout like eGFP, which likely explains the observations from these additional experiments. The main point is that one can't compare performance of switches (especially fold changes in activity) characterized in one assay system (e.g., transcription factor + enzymatic reporter) to another (e.g., eGFP). Since most of their other switches are characterized in the eGFP reporter system, it seems reasonable that the activity of the neomycin switches should be also reported in this same reporter system in the main manuscript text to make this point in difference between assay systems more clear (along with the Gal4 enzymatic reporter system if the authors so decide).

Separate comments:

- The authors are varying the threshold level of calling the response of their logic gates quite substantially. For example, it's not clear that the threshold set for the OR and NAND gates are well supported given the intermediate values for the single-input activities (Figure 3c). In addition, the difference in the histograms between the 0 and 1 states are quite small (as shown in Figure 3). Given other more recent RNA-based two input switches and genetic logic gates in general, it is not well supported that the twister ribozyme is giving improved performance in this context and the calling of some of the behavior of the switches (particularly OR and NAND) does not appear to be well supported from the data.

Reviewer #3 (Remarks to the Author):

The authors have addressed most of my concerns and the manuscript may be suitable for publication in Nat Comm.

Response to the comments of Reviewer 2:

Reviewer #2 (Remarks to the Author):

The authors have addressed most of our remaining comments. However, their findings and results for the third comment are notable. Specifically, their findings clearly show that switches will have different performances when characterized in different reporter systems (i.e., a direct fluorescent readout (GFP) versus an indirect enzymatic reporter readout (B-gal)). These results highlight the point that it is difficult (if not impossible) to make performance comparisons across papers that characterize switches in different assays, systems, etc; and that direct comparisons require that the switches be characterized in the same exact assay in the same experiment.

It's expected that enzymatic assays will be more sensitive and thus provide greater sensitivity to smaller activity readings (and greater reported dynamic ranges) than a direct fluorescent readout like eGFP, which likely explains the observations from these additional experiments. The main point is that one can't compare performance of switches (especially fold changes in activity) characterized in one assay system (e.g., transcription factor + enzymatic reporter) to another (e.g., eGFP). Since most of their other switches are characterized in the eGFP reporter system, it seems reasonable that the activity of the neomycin switches should be also reported in this same reporter system in the main manuscript text to make this point in difference between assay systems more clear (along with the Gal4 enzymatic reporter system if the authors so decide).

Response: *We agree with reviewer 2 that performance comparisons are "...difficult (if not impossible)...". This is why a comparison of eGFP results of a neomycin switch in yeast to switches responding to theophylline and TPP in E. coli is not very helpful. We have explicitly stated that different performances could result utilizing different reporter and expression setups, see Supporting Note 5. We have now added a similar paragraph to the main text (discussion) in order to further address this issue:*

"Artificial aptazyme-based riboswitches located in the 3'-UTR of the target gene were already developed in yeast by other groups^{3, 11-13}. The first ribozyme-based switches in yeast developed by Smolke and coworkers^{11, 13} were shown to have switching performances around 2-2.5-fold¹⁴. The rationally designed hammerhead (HHR)-based tetracycline riboswitches of Sues and coworkers showed switching performances up to 2.5-fold. In these works the switching performances were quantified via direct detection of the gene product (a fluorescent protein). In 2014 our group published an in vivo selection method for the generation of different HHR-based neomycin riboswitches in yeast, with switching performances up to 25-fold³. However, in the latter work, the switching performances of the in vivo selected riboswitches were quantified using the same GAL4-LacZ system as described in the present work (see Supplementary Note 1). For this reason a direct comparison of the switching performances of our neomycin switches to the ones previously published is not possible. We cannot exclude the possibility that our indirect approach of controlling reporter gene expression via control of the expression of a transcription factor possesses an amplification effect. In this case, the reported

ribozyme-based switches would work less efficiently when employed in a direct setup (i.e. inserted immediately into the 3'-UTR of the reporter gene)."

In addition, the reviewer states: "The main point is that one can't compare performance of switches (especially fold changes in activity) ...". It is important to note that we did not compare performances at all. In fact, all performances (i.e. fold changes) had been already removed from the main figures in response to the reviewers comments.

Separate comments:

- The authors are varying the threshold level of calling the response of their logic gates quite substantially. For example, it's not clear that the threshold set for the OR and NAND gates are well supported given the intermediate values for the single-input activities (Figure 3c). In addition, the difference in the histograms between the 0 and 1 states are quite small (as shown in Figure 3). Given other more recent RNA-based two input switches and genetic logic gates in general, it is not well supported that the twister ribozyme is giving improved performance in this context and the calling of some of the behavior of the switches (particularly OR and NAND) does not appear to be well supported from the data.

Response: *We agree with the reviewer that a more digital response to the two mentioned switches would be advantageous. However, as detailed in earlier responses, in the end we report two-input switches that respond to the presence of the two ligands with changed gene expression. The mentioned switches respond with significantly changed expression outputs under the different conditions as demonstrated with bulk fluorescence and luminescence as well as flow cytometry measurements. This issue is discussed in the Supporting Note 6.*